# An Exploratory Network Analysis of Discussion Topics About Autism Across Subreddit Communities

**DOI:** 10.3390/bs15060812

**Published:** 2025-06-13

**Authors:** Skylar DeWitt, Kendall Mills, Adam M. Briggs

**Affiliations:** Department of Psychology, Eastern Michigan University, Ypsilanti, MI 48197, USA; sdewitt5@emich.edu (S.D.); kmills17@emich.edu (K.M.)

**Keywords:** autism, topic modeling, machine learning, network analysis, neurodiversity, social media

## Abstract

Using an inductive computational approach, our present data exploration sought to use machine learning methodology to define and identify patterns and gain insight into autism-related discussions on Reddit across three different categories of subreddits: (a) individuals who self-identify as autistic, (b) parents of individuals on the autism spectrum, and (c) behavior therapists. By doing so, we sought to review authentic autism-related discussions and identify important topics that emerged across these three demographic groups, including insights related to assessing and treating challenging behavior. Following basic and advanced preprocessing, our extraction resulted in 57 subreddits and 46,914 comments from autism spectrum subreddit members, 46 subreddits and 27,838 comments from parent subreddit members, and six subreddits with 3163 comments from behavior therapist subreddit members. Subsequent network analyses revealed interesting patterns of discussion within and across subreddit groups that may be used to inform support and resources, practice considerations, and future directions for research.

## 1. Introduction

Autism is a developmental disability that affects nearly every aspect of an individual’s lived experience (e.g., executive functioning, sensory processing, motor skills, communication, social behavior, adaptive functioning; [4]). Challenges that often accompany autism include difficulties with social or emotional reciprocity, forming or maintaining relationships, non-verbal communication skills, inflexibility or insistence on sameness, and addressing the emergence of maladaptive challenging behavior, which range from mild to severe ([2]). In many cases, adequate support for autistic individuals requires a combination of self-advocacy, parent/caregiver involvement, and intervention planning on behalf of trained mental health practitioners (e.g., behavior therapists). To improve our ability to provide support for these individuals, it is essential to consider input and perspectives from all three sources to identify and address gaps in service delivery (e.g., unmet needs).

Qualitative data collection methods (e.g., extended interviews, focus groups) can be used to develop an in-depth understanding of an issue because these methods generally allow for nuanced responses and the opportunity for unanticipated insights through the use of open-ended questions that quantitative methods may miss ([17]). Recently, qualitative strategies have been used to understand the perspectives of autistic individuals as they pertain to their (a) experiences and (b) support for common therapeutic interventions. For example, [13] ([13]) performed an inductive qualitative analysis to gather 32 autistic adults’ perspectives and experiences related to ‘stimming’ (i.e., stereotypic behavior, a core autism symptom) via a combination of in-depth interviews and focus group methodologies. A manual thematic analysis of the data highlighted the perceived importance of stimming as a functional and adaptive self-soothing mechanism. These findings provided insight into the participants’ reasons for objecting to a common behavioral treatment designed to eliminate the behavior. As a result, the authors discussed the potential need for evidence-based interventions that foster the development of non-harmful (i.e., non-disruptive and non-injurious) stimming rather than eliminating the behavior altogether. Similar research has been conducted to understand unmet treatment needs and barriers to treatment from the perspective of caregivers of autistic individuals ([16]), as well as to better understand the role of caseload management in burnout experienced by Board Certified Behavior Analysts (BCBAs), which may function as a barrier to adequate service delivery ([15]).

While the aforementioned qualitative studies highlight perspectives and unique unmet needs from each respective population of individuals, noteworthy limitations exist. First, qualitative research requires participants to describe their thoughts and feelings to the research staff. Such an approach may be susceptible to social desirability bias or a tendency to answer questions in a manner viewed as socially acceptable ([7]). This is problematic because it may limit the authenticity of the data collected, as participants may refrain from sharing aspects of their perspective due to fear that such responses will be viewed unfavorably by the researchers leading the interview or focus group. Second, due to the wealth of data produced by qualitative research methods, the sample size of such studies is often relatively small and usually limited to 25–30 participants to prevent researchers from becoming inundated with more information than they can reasonably sift through via manual thematic analysis ([9]).

One way that qualitative data collection methodology has recently been expanded upon to mitigate some limitations involves leveraging data science to make sense of large collections of text data (e.g., social media data; [28]). Techniques utilizing large collections of text data (often described as ‘big data’) from social media have been implemented to explore online discussions across various topics and demographics. For example, [23] ([23]) investigated whether unsupervised topic modeling with Latent Dirichlet Allocation (LDA) generated interpretable psychological themes that could be used to predict outcomes of clinical assessments of depression and neuroticism. This study used Pennebaker’s Linguistic Inquiry and Word Count lexicon to analyze 6459 college students’ essays. They determined that the model produced relevant and population-specific themes related to depression and neuroticism, which correlated with students’ scores on clinical measures of these features, thereby serving as a helpful supplement to formal clinical assessments of depression and neuroticism. Thus, big data analysis may be a fruitful avenue for conducting a thematic analysis of clinically relevant individual perspectives from autistic individuals on a larger scale than possible in the Kapp et al. study. Similarly, big data may help us explore the needs of caregivers of children and adults with autism and their behavior therapists.

This preliminary data exploration sought to use an inductive computational approach to identify and define topics of conversation about autism that might offer potential insights into the perspectives and needs of users that make up three distinct categories of subreddits: (a) self-identified (self-ID) autistic individuals, (b) parents of autistic individuals, and (c) behavior therapists. We hope this preliminary exploration may quantify the number and nature of conversation topics in these communities and highlight their relevance to the assessment and treatment of autistic individuals. Doing so may reveal important themes that can, through more targeted exploration, potentially inform how professionals may better understand and improve their coordination, planning, and support of these communities throughout the therapeutic process.

## 2. Method

### 2.1. Materials

Several social media sites host users from our desired demographic; however, one social media site that is particularly well-suited for our use in exploring these questions is Reddit. Reddit comprises discussion threads called subreddits, which are communities of people who identify similarly in some way (e.g., hobbies, personality, lifestyle, diagnosis). In particular, there are individual communities populated with (a) users who self-ID as having autism, (b) parents of autistic individuals, and (c) behavior therapists who serve in clinical settings with individuals with autism (henceforth referred to as the “autism”, “parent”, and “behavior therapist” groups, respectively).

We performed all procedures and analyses in R (Version 4.2.2; [21]). R is a statistical software that allows users to authenticate with application programming interfaces (APIs), extract data from social media websites, and analyze the subsequent data with user-created packages. For this analysis, we utilized the package RedditExtratoR ([25]) to gather subreddits relevant to our three targeted demographic groups. This package allowed us to access up to 500 comments per Reddit post directly through Reddit’s API. We then utilized the package ‘stm’ ([26]) to generate topic models and network analyses.

### 2.2. Data Extraction

We extracted data from subreddits demarcated by their relevance to our three interest groups. These subreddits were initially identified by utilizing Reddit’s search engine and examining relevant subreddits. For the autism subreddits, the following searches were made in Reddit’s search engine: “autism”, “asd”, “Asperger’s”, and “autistic”. For parents, the following searches were made: “parents”, “mom”, “dad”, “autism parents”, “stepparents”, “stepmom”, “stepdad”, “adoptive parents”, and “foster parents”. For the behavior therapist subreddits, the following searches were done: “behavior analysts”, “behavior analysis”, “aba”, “applied behavior analysis”, “bcba”, “rbt”, and “behavior technician”. We conducted a broad search using these terms to ensure all possible subreddits that were good candidates for analysis were captured.

Once each of these searches was performed, each subreddit search result was inspected to determine whether it met the following criteria for inclusion in this data exploration: (a) subreddit communities specify they exist as a space for our specific populations of interest in their description or are otherwise indicated by visually inspecting the communities and ensuring posts were made by and for our targeted demographic; and (b) posts consist of questions, shared experiences, and opinion posts relevant to the community of interest. Subreddits were discarded if they centered around conversations about specific media or persons.

After identifying these subreddits, the package ‘RedditExtractoR’ was used to scrape Reddit’s API and access comments on these subreddits ([25]). A keyword search of “autism” was applied to ensure conversations in each subreddit concerned autism. The post text from the original poster was not included in the analyses. Data retrieval was limited to the last five years (i.e., 2017–2022) to capture more current discussion topics. The resulting comments were exported as a separate Excel file for each community. These files were separated for distinct analysis to ensure that the data from the smallest group (behavior therapists) would not be drowned out by the data from the largest group (autistic individuals). We took this measure in line with previous research suggesting that separating demographic groups into separate corpora can prevent data salient to smaller groups from being overshadowed by the proportion of comments in larger groups ([34]; [39]).

### 2.3. Data Cleaning

Once these Excel files were created, they were imported back into R for cleaning. First, corpora (i.e., a collection of texts) were created using the package ‘stm’ ([26]). Here, we initialized basic preprocessing measures, such as removing html tags, stop words, bot comments, and common initialisms (e.g., “omg”, “lol”, “ttyl”). The package ‘textstem’ ([24]) assisted with advanced preprocessing measures, such as giving dictionary meanings to words and converting them to their root form (i.e., lemmatization). After data cleaning, the corpora vocabulary, metadata (i.e., data that provide information about other data), and comments were prepped for topic modeling using the ‘stm’ package. As a result, the final cleaned corpora comprised 46,914 comments in the autism subreddits, 27,838 comments in the parent subreddits, and 3163 comments in the behavior therapist subreddits.

### 2.4. Topic Modeling

We used Latent Dirichlet Allocation (LDA), an unsupervised machine learning technique that uses probabilistic modeling, to implement topic modeling. In this method, each comment is assumed to consist of a different proportion of topics. A crucial step in determining the best LDA model output is to find the optimal number for K (i.e., the number of topics). There are several validated methods for selecting K in topic modeling; however, because of the large number of small comments, we chose to follow the previous recommendation of focusing on semantic coherence (how often words within a topic co-occur) and exclusivity (the uniqueness of words to each topic) from [27] ([27]). Of the models with the highest semantic coherence and exclusivity, the models that had higher held-out likelihood and lower residuals were selected. Held-out likelihood refers to the model’s predictive value (i.e., if some comments were held out from the topic modeling process, would it still generate similar results). Residuals refer to statistical error. Thus, this iterative process was completed for each model until scores that met these criteria of higher held-out likelihood and lower residuals were obtained.

We labeled topics using simplified frequency-exclusivity (FREX) scoring in conjunction with human judgment. This method summarizes words with the harmonic mean of the probability of appearance under a topic and the exclusivity to that topic. FREX words have effectively provided more semantically intuitive representations of topics ([27]). Once the FREX words were identified, each topic’s top 10 Reddit comments were printed. We then manually labeled topic names using the FREX metric and our judgment to inform our decisions. This process entailed reviewing the top 10 representative comments and associated FREX words. Independent topic labels were assigned, and agreement was confirmed through comparison.

Raters agreed on 66 out of 71 topics (92.96%) across network analyses. For topics in which raters identified disparate labels, raters combined the individual labels to create a more comprehensive label that captured the essential elements of both raters’ observations. For example, Topic 30 in the autism subreddit network analysis, “Worldbuilding”, was initially labeled as “Media and Religion” and “Theology” by the raters. To address this discrepancy, we printed 10 additional comments, discussed the results, and concluded that a proper through-line for comments concerning medieval video games and anime, contemporary and classic fantasy-themed media, and ancient religious and philosophical stories was “Worldbuilding”.

### 2.5. Network Analyses

A network analysis was performed to plot a network in which nodes (i.e., the topics of interest) and edges (i.e., the connections between nodes) indicate a positive correlation, such that positively correlated nodes are connected by a line, and the thickness of the line indicates the strength of the correlation. Correlations reflect the frequency with which topics were found within the same comments. Using the package stm ([26]), tidygraph ([19]), and ggrpah ([18]), a weighted, undirected network model was produced for each group, where nodes represent topics, and lines linking the nodes represent the correlation between topics. Additionally, we implemented a community graph partitioning algorithm to visually arrange topics by community membership (i.e., which topics share content similarities). Specifically, we applied a *walktrap* algorithm ([20]) using tidygraph ([19]), a hierarchical clustering algorithm based on random walks (a mathematical process for modeling the probability that two nodes are related to one another). This algorithm assumes that if you perform random walks on the graph, the walks are more likely to stay within the same community, as only a few edges lead outside a given community.

### 2.6. Topic Proportions

To visualize the proportions of individual topics, we extracted the stm topic models Theta (Θ). This allowed us to visualize the probability of a comment belonging to a particular topic. Each topic’s Theta was extracted from the topic model using the package stm ([26]). Results were then plotted using ggplot2 ([37]) to rank the topics from those that made up the greatest expected proportion of comments to those that made up the fewest expected proportion of comments. Essentially, this serves as a metric for which topics were discussed the most frequently throughout the subreddits captured by our data pool.

Our final topic model for the autism subreddits included 57 subreddits, 46,914 comments, and 32 unique topics. Second, our parent topic model consisted of 46 subreddits, 27,838 comments, and 22 unique topics. Finally, our behavior therapist topic model consisted of six subreddits, 3163 comments from behavior therapist subreddit members, and 16 unique topics.

## 3. Results

### Topic Proportions

We first calculated the anticipated proportions of topics to identify which ones constituted the highest expected proportion within the comments. First, for our autism topic model, expected topic proportions ranged from 0.02 to 0.06 (Figure 1). The topic that comprised the greatest proportion of comments and had the greatest expected topic proportion for the autism subreddits was “Animal Facts” (expected topic proportion = 0.06). This topic consisted of users sharing detailed information about types of animals, animal genetics, and favorite types of animals. The topic that made up the lowest expected topic proportion in the autism model was “Social Systems” (expected topic proportion = 0.02), which contained conversations about the social behavior of humans and other animals.

Next, in our parent topic model, expected topic proportions ranged from 0.03 to 0.06 (Figure 2). The topic that made up the greatest proportion of comments and had the greatest expected topic proportion for the autism subreddits was “Vaccines” (expected topic proportion = 0.06). This topic consisted of users discussing the link between autism and vaccines, engaging in discourse about the anti-vax movement, and stating support for or against vaccines. The topic that made up the lowest expected topic proportion in the autism model was “Navigating Marriage” (expected topic proportion = 0.03), which contained conversations about divorce, marital strain, and giving advice to and receiving advice from others on approaching difficult conversations with partners.

Finally, our behavior therapist topic model consisted of expected topic proportions ranging from 0.04 to 0.09 (Figure 3). The topic that comprised the greatest proportion of comments and had the greatest expected topic proportion for the autism subreddits was “Client and Practitioner Barriers” (expected topic proportion = 0.09). This topic consisted of users discussing various barriers to successful treatment or practice (e.g., cultural/language barriers with parents, a lack of adequate training). The topic that made up the lowest expected topic proportion in the autism model was “Defining Autism” (expected topic proportion = 0.04), which contained conversations about the diagnostic criteria for autism, the neurological bases of autism, and the etiology of autism.

## 4. Network Analysis

Across autism subreddits, community membership partitioning detected four unique communities (Figure 4), which were numbered arbitrarily. Community 1 comprised 16 topics primarily focusing on social behavior, self, others, and their consequences in personal relationships and broader social contexts. Community 2 comprised eight topics, with overarching themes concerning testing, certifications, qualifications, and ability. Community 3 comprised five topics, which belonged to themes related to sensory experiences. Community 4 comprised three topics concerning specific and broad special interests.

Community membership partitioning across the parent subreddit communities detected five unique communities (Figure 5). Community 1 comprised seven distinct topics, primarily focusing on testing, treatment, foster care, feelings of judgment, and challenges associated with raising an autistic child. Community 2 comprised seven unique topics, generally belonging to overarching themes related to sharing experiences with parenting. Community 3 comprised three topics concerning identifying a child’s abilities/needs and developing a plan to support those needs. Community 4 comprised three topics, primarily concerned with parents sharing experiences of their child’s unique behaviors and preferences. Finally, Community 5 comprised two topics, “vaccines” and “prenatal causes of autism”, which included conversations surrounding debates about how autism is caused.

Community membership partitioning across the behavior therapist subreddit communities detected three unique communities (Figure 6). Community 1 comprised six topics related to an overarching theme of definitions, assessments, and ethics. Community 2 comprised five topics related to general themes concerning job roles, duties, and stressful experiences. Finally, Community 3 comprised four topics related to interpersonal relationships and reasons for leaving their therapeutic field.

## 5. Discussion

Although research in this area is preliminary, machine learning shows promise for psychological research, particularly in autism research, where it could aid in collecting and quantifying the experiences of autistic individuals, their parents, and their behavior therapists. In particular, to investigate the perspectives of these three groups, our study utilized machine learning methodology to explore topics of conversation surrounding autism across these three demographic groups and highlight their relevance to the assessment and treatment of autistic individuals.

We found that the autism subreddits discussed a much broader range of topics than the other two subreddit categories, with more community-building and intimate interpersonal discussion than the parent and behavior therapist subreddits. Although parents discussed a range of topics, most topics of conversation centered around seeking advice or sharing difficult experiences related to being a parent (e.g., enforcing bedtime routines). The behavior therapist subreddits appeared to discuss the narrowest range of topics, with most conversations directly related to users’ job responsibilities (e.g., navigating issues with insurance billing). An interesting interpretation of these results is the affirmation that autism is an identity that impacts every aspect of an individual’s life (e.g., school experiences, personal interests, experiencing discrimination, co-occurring conditions, sensory processing). In contrast, parenthood is a social/familial role, and being a behavior therapist is an employment role. In other words, a person’s autism is relevant across contexts because it impacts how they interact with the world around them in myriad ways.

Meanwhile, parenthood or a person’s job title is relatively less relevant outside their respective contexts (e.g., whether someone is a parent or a behavior therapist does not directly impact their personal interests or how they process sensory information). Thus, it makes sense that the conversation topics in these subreddit communities reflect this. Thinking about autism in this way (i.e., as a facet of identity with relevance that extends beyond challenges with social–emotional reciprocity) provides unique insight into the interests, values, and social experience of an autistic individual on a discussion-based platform, which may have important implications for clinical practice. For instance, gaining insights from autistic individuals about how they experience the world could inform specific behavioral targets that align with their values and preferred approaches toward assessment and treatment, which could be identified and incorporated into practice. Further, this information may also highlight the need to connect autistic individuals with various services (e.g., behavioral, occupational, academic, vocational) that address concerns and meet desired goals across many aspects of their lives. For example, a behavior therapist may become aware of a client’s interest in working with cars and, in response to this, connect them with vocational support programs that specialize in working with autistic adults to help them obtain the skills and educational/vocational training required to become a mechanic (e.g., certificate programs, apprenticeships).

Other interesting outcomes from this study concern the distinct clusters of topics within each of the three subreddit groups. In the autism subreddits, we found a pattern in which topics related to sensory experiences (Sensory Sensitivities, Auditory Sensations) and highly focused interests (Animal Facts, Worldbuilding, and a general “Special Interests” topic) shared community membership with a topic describing and celebrating autistic representation in media (Autistic Fictional Characters). Interestingly, these topics did not share community membership with topics related to adverse social outcomes (e.g., Shame or Social Mistakes). This lack of connection to adverse topics may suggest that the unique sensory experiences and heightened capacity for specialized interests of autistic individuals may not be experienced by the individual as socially impairing features of autism. Although these characteristics correspond to well-documented diagnostic features of autism (e.g., “highly restricted, fixated interests that are abnormal in intensity or focus”, as described by the *Diagnostic and Statistical Manual of Mental Disorders—5th Edition*), this finding may suggest that unequivocally interpreting the presence of specialized interests through the framework of a deficit model (i.e., a focus on what autistic individuals lack rather than what they naturally excel at or gravitate towards; [3]), in which the indicated treatment may be to expand the individuals’ repertoire of interests, may not be warranted for all individuals. Instead, for individuals not experiencing distress or functional impairment due to a limited repertoire of interests, these interests may function as a source of social bonding between autistic individuals, which should be promoted. This is consistent with findings from existing literature that outline how special interests can act as a signature strength in autistic individuals who cultivate positive emotions and coping (e.g., [35]). Hence, future studies might investigate the social and clinical implications of empowering individuals to utilize their unique interests to connect with other autistic individuals who share related interests (e.g., establishing clubs or hobby groups that may function as a source of social reinforcement).

Interesting patterns we observed in the parent subreddits included many topics dedicated to seeking advice, offering advice and support, and requesting guidance in navigating available supports for their autistic child(ren). In general, these themes suggest that this community may be experiencing a lack of support or access to other resources in their daily lives, as their discussions frequently highlighted barriers to parenting autistic children (e.g., understanding sensory differences, obtaining diagnostic assessment), co-parenting, and navigating relationships with one’s partner. This aligns with previous research highlighting parents’ interest in obtaining relevant support/resources while on waitlists for services (e.g., strategies for teaching and maintaining a child’s skills, support for managing behavioral challenges, mental health services, advocacy support, respite care; [33]). Specifically, another interesting theme was that parents frequently asked other users for input as to whether their child’s behavior or developmental trajectory was suggestive of autism, which may speak to a need for resources or parent training on these topics. For example, these resources may include brochures, flyers, or printed summaries of academic literature outlining early signs of autism and contact information for local diagnostic clinics, which pediatricians or other general practice and family doctors may provide. Potential parent training might cover general, evidence-based behavior management and early learning/skill acquisition strategies. It should be delivered by licensed professionals (e.g., physicians and psychologists) so that parents feel less need to seek informal support from strangers through various social media outlets. These trainings could be disseminated as free online video modules (e.g., via YouTube or available on the Autism Speaks website) so parents can access them conveniently. Further, it may be advantageous to offer other educational resources that support caregivers’ understanding of the process and intricacies of the assessment and treatment progression so they can anticipate this sequence and be prepared for their role throughout and following intervention.

Lastly, as previous research has suggested ([6]; [22]), information about autism and resources for parents online is often misleading, a concern possibly reflected by Community 5 (Figure 2), which included topics of Vaccines and Prenatal Causes of Autism. Aligned with current social network data literature predicting the dominance of vaccine misinformation on social media platforms in the next decade without intervention ([12]), our analyses emphasize the necessity for evidence-based interventions against the potential spread of vaccine misinformation in parent subreddit communities. Future research may consider exploring applications of misinformation control measures that have previously demonstrated success, such as preventive actions (e.g., prompting critical reasoning skills; [36]) or the use of a manual (human) or automatic (artificial intelligence) moderator to actively flag misinformation and provide correct information within the context of an online forum ([10]).

Finally, our analysis of the behavior therapist subreddits revealed a primary theme of users articulating complaints about systemic issues within the field of behavior analysis (e.g., lack of adequate training or supervision, challenges with insurance billing, feeling underpaid given the work they perform), which drew particular attention to topics related to burnout and stress consistent with the existing literature (e.g., [14]). Possible negative implications of this observation may include, but are not limited to, venting without receiving any tangible solutions, increased likelihood of turnover, and behavior therapists leaving the profession entirely ([31]). The field of behavior analysis may respond to these data by engaging in efforts to prevent burnout at the source by directly addressing these grievances, such as improving training and supervision ([8]), providing more administrative support from employers ([31]), and increasing wages based on current market rates ([32]). In addition, several topics revealed discussion surrounding behavioral assessment, sharing treatment recommendations, and the ethics of treatment implementation. Monitoring these discussion topics could identify common questions, concerns, and misunderstandings therapists may have related to assessing and treating challenging behavior and could indicate where more experience, training, education, and resources are needed to ensure therapists are aware of best practices and have the required skills, experience, and support to incorporate these evidence-based procedures into their practice to assess and treat challenging behavior effectively.

### 5.1. Limitations

Our model has a few noteworthy limitations. Our topic model method involved an LDA algorithm, which is limited by its unsupervised nature. We could not adequately capture the comments’ sentiment or linguistic structure when performing analyses. Therefore, while the topic labels may suggest different tones and sentiments, we did not perform any direct sentiment analysis. As such, our examples of positive or negative tone/sentiment come from our interpretations while reading metadata samples and not from formal statistical applications. Second, our analysis cannot conclude whether the information reported in these communities accurately reflects actionable behavior (i.e., whether a personal anecdote presented by a user actually occurred). As a result, our conclusions should be interpreted in a speculative rather than definitive manner. Next, Reddit’s API allows individuals to scrape up to 500 comments per subreddit; thus, this study captured only a subset of comments posted between 2017 and 2022. As such, the present study does not reflect a comprehensive evaluation of all communications within relevant subreddits during this five-year period.

It is also important to note that Reddit generally represents a specific demographic. Specifically, the reported demographics of the Reddit user base consist of users who are 70% White, and nearly half come from the United States ([30]). Furthermore, while we sought out specific subreddits that represented our targeted demographic groups, it cannot be said with certainty that these subreddits contain comments solely from our targeted demographics. As such, whether or not the comments were directly linked to someone who identifies as an autistic individual, a parent of an autistic individual, or a behavior therapist is unknown. Finally, the number of comments we could extract from the behavior therapist subreddits was substantially smaller than the other two subreddits. As of 3 April 2023, a total of 61,337 individuals hold a BCBA certification, 5520 individuals hold a BCaBA certification, and 136,113 individuals hold a Registered Behavior Technician (RBT) certification ([5]). We do not have data on the number of users whose comments contributed to the present study; however, given that we only obtained 3163 comments from the behavior therapist subreddits, it can be assumed that our current sample represents only a tiny fraction of this community.

### 5.2. Future Directions

Several future directions for research can be inferred from the outcomes of this study. First, it is likely impossible to meaningfully describe and analyze every topic and connection between topics uncovered by the present data pool within the confines of one paper. Thus, it is recommended that future researchers take a more targeted approach in similar future analyses using insight and inspiration from this broad exploratory analysis to deeply examine the nuances of topics relevant to the assessment and treatment of autistic individuals within these communities.

For example, our autism network analysis highlights the need to consider autistic perspectives in developing socially valid support for autistic individuals. We chose to describe the potential implications of conceptualizing special interests characteristic of autism as a signature strength that may be used as a foundation for designing interventions that foster social connection. However, we made this decision at the expense of expounding upon other areas of this network analysis that may be of broader interest to researchers and practitioners. Future researchers may instead wish to perform a similar analysis of the content of discussions within the Gender and Sexuality topic. This may be of key interest because, when compared to the general population, autistic individuals are more likely to be transgender, non-binary, and/or non-heterosexual ([29]). Thus, there may often be reasons to consider factors related to gender and sexuality when designing supports for autistic individuals (e.g., navigating discrimination from multiple angles), making more education, support, and research on this topic a warranted endeavor.

Further, our network analysis for parents of autistic individuals may provide insight into how practitioners may modify behavior therapy protocols to reduce caregiver barriers to treatment adherence. Although there is an emerging literature on understanding these caregiver barriers in telehealth and in-person service delivery models (e.g., [1]; [11]; [38]), social media forums such as Reddit may be another valuable way to gather data about the barriers parents face, which could be used to inform treatment recommendations or strategies to train parents on how to implement behavioral therapies with their children in a home setting. Furthermore, it may be beneficial for those working in the field of behavior analysis to connect parents with others in their community (e.g., via local or online parenting groups that provide evidence-based guidance), as this may facilitate the formation of long-term supportive relationships.

Finally, it may be worthwhile to conduct sentiment analyses on traditionally divisive topics identified during the present study (e.g., vaccines) to understand whether and to what extent professionals with expertise in autism may need to anticipate misinformed opinions on the part of parents. Additionally, these procedures could be utilized to conduct a more thorough exploration of sources of burnout among BCBAs and RBTs to inform more detailed recommendations for retaining these employees within their organizations and promoting greater job satisfaction. Finally, assessing gaps in therapist knowledge and barriers to best-practice implementation related to assessing and treating challenging behavior may identify where educational programs and supervised clinical settings may improve training experiences so that behavior therapists are better prepared to address their clients’ challenging behavior in practice.

## Figures and Tables

**Figure 1 behavsci-15-00812-f001:**
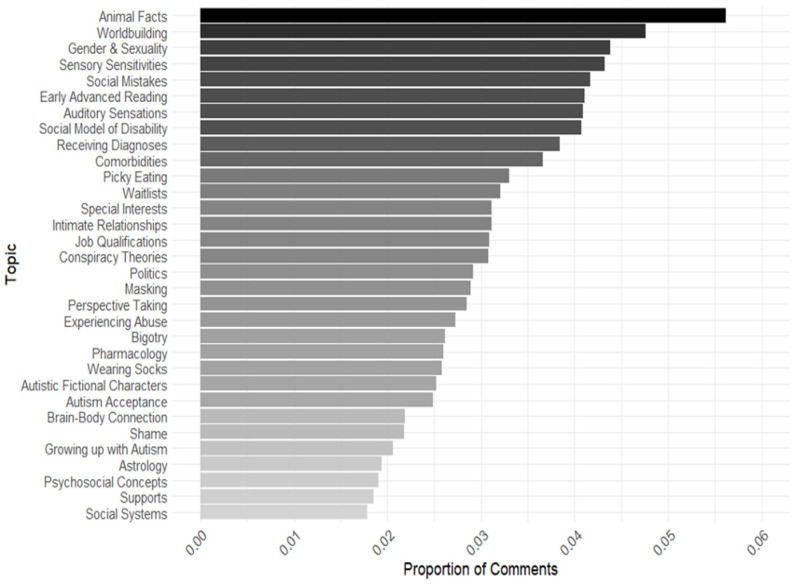
Topic proportions for autism subreddit group. Note: This graph represents the topics that made up the greatest expected proportion of comments in the autism subreddits. Theta (Θ) was extracted from its respective topic model to plot expected topic proportions.

**Figure 2 behavsci-15-00812-f002:**
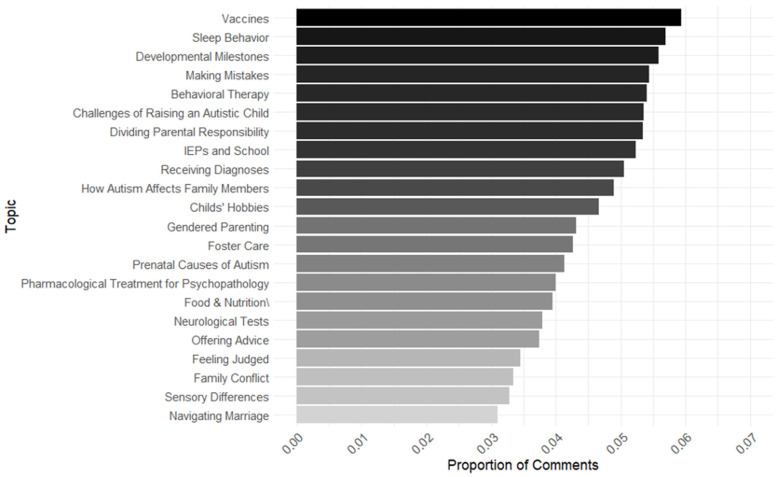
Topic proportions for parent subreddit group. Note: This graph represents the topics that made up the greatest expected proportion of comments in the parent subreddits. Theta (Θ) was extracted from its respective topic model to plot expected topic proportions.

**Figure 3 behavsci-15-00812-f003:**
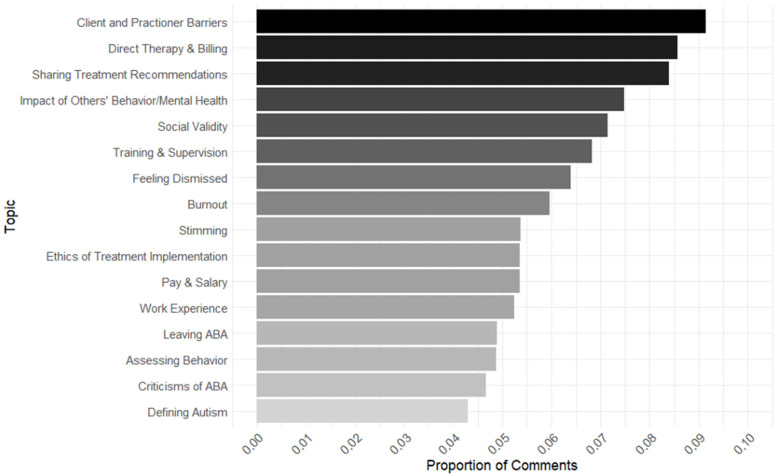
Topic proportions for behavior therapist subreddit group. Note: This graph represents the topics that made up the greatest expected proportion of comments in the behavior therapist subreddits. Each topic’s Theta (Θ) was extracted from its respective topic model to plot expected topic proportions.

**Figure 4 behavsci-15-00812-f004:**
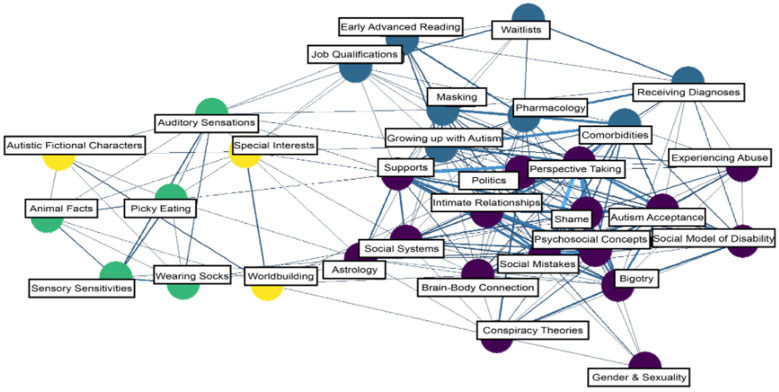
Topic model for autism subreddit group. Note: Nodes (circles) are topics, and edges (i.e., the connections between nodes) indicate a positive correlation. The thicker the edge line, the stronger the correlation between topics.

**Figure 5 behavsci-15-00812-f005:**
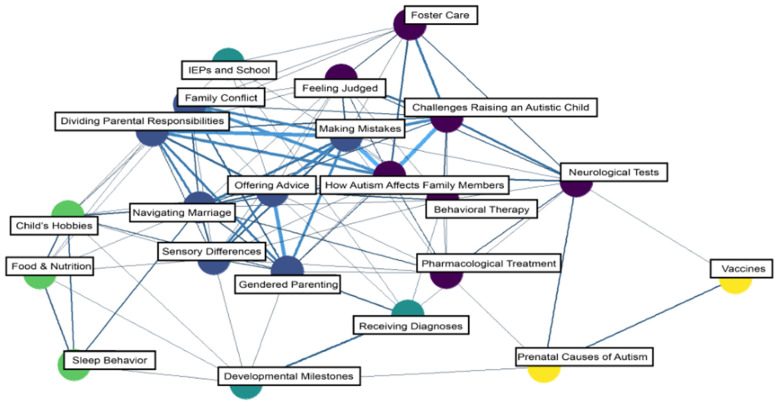
Network model of topics: parent subreddits. Note: Nodes (circles) are topics, and edges (i.e., the connections between nodes) indicate a positive correlation. The thicker the edge line, the stronger the correlation between topics.

**Figure 6 behavsci-15-00812-f006:**
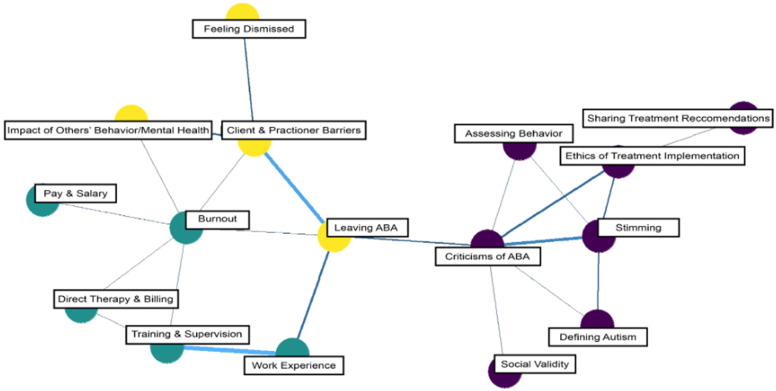
Network model of topics: behavior therapist subreddits. Note: Nodes (circles) are topics, and edges (i.e., the connections between nodes) indicate a positive correlation. The thicker the edge line, the stronger the correlation between topics.

## Data Availability

Data are available upon reasonable request from the corresponding author.

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
