# Peer review of "An Exploratory Network Analysis of Discussion Topics About Autism Across Subreddit Communities"

_behavsci, 2025, doi:10.3390/bs15060812_

Round 1
Reviewer 1 Report
Comments and Suggestions for Authors
I am thankful for the opportunity to review, “An Exploratory Network Analysis of Discussion Topics About 2 Autism Across Subreddit Communities” for Behavioral Sciences. The purpose of the paper was to describe topics of conversation amongst autistic individuals, parents of autistic individuals, and clinicians providing behavioral therapy to autistic individuals. The authors pursued this aim by using machine-learning to gather and organize comments from subreddits related to each of their target demo graphic groups. Their results provided common topics of discussion for each group in addition to quantifying and categorizing covariance between individual topics. The authors suggest that such a methodology could provide insight into the needs, preferences, facilitators, and barriers experienced by members of each of these groups. Moreover, such an approach makes it possible to do this at a scale that would not be feasible using less computationally intensive qualitative methods.
As a behavior analyst who frequently conducts clinical research with autistic children and has dabbled in computationally intensive methods of data analysis, I was very intrigued by the premise of this study. I enjoyed reading the manuscript and feel that I learned quite a bit, given that this is somewhat afield from my area of expertise. I appreciated the comprehensive nature of the authors approach and their points about the potential utility of sampling opinions from these three interrelated. Although, I have several suggestions for improvement and would like to see some additional detail and data, I feel that the authors can readily address these issues through working with the handling editor. Therefore, I am happy to recommend that the manuscript be accepted, pending minor revisions. My specific suggestions for improvement are provided below.
Introduction
- Line 31: It may be helpful to break down the diagnostic criteria into social and RRB domains and to presenting each component with a similar syntax (e.g., change “addressing the emergence of challenging behavior”)
- Line 35: Is mental health practitioners the most appropriate general label?
- Line 46: Although such terminology is pervasive, the authors may consider including terms like behavioral difference or tendency rather than symptom (especially given the goal of the paper).
- Line 84: Rather than referencing Kapp et al specifically, it may be clearer to say previous qualitative work related to experiences of and supports for autistic individuals.
- In the introduction, the authors could do more to develop rationale for their use of subreddits, specifically. Specifically, it may be worth discussing whether or not certain communities may introduce bias in terms of what is said, how it I said, and by whom (i.e., why should we expect these samples to be sufficiently representative?).
- Line 75: Details like unsupervised modeling, Latent Dirichlet Allocation, and Pennebaker's Linguistic Inquiry seem sufficiently complex to merit elaboration. Alternatively, if they are not essential to demonstrating the utility of this general approach, then they could be removed.
- Line 93: Given the goal of the paper, it may be better to avoid “treatment of autistic individuals” and instead describe strategies for supporting members of this population or, at least, treating/intervening on their behavior.
Method:
- Line 110: Missing a “c” in RedditExctactoR
- Line 119: The search terms for subreddits relevant to parents of autistic individuals seem especially general and likely to return many irrelevant results. Relatedly, the search terms are centric to behavior analysis seem more specific than the target population of “behavior therapists who serve in clinical settings with individuals with autism”.
- Given the fundamental importance of identifying relevant and excluding irrelevant subreddits it would seem important to evaluate interrater agreement for application of the inclusion criteria. This is especially important given the subjective nature of the criteria. Was interrater agreement obtained. If so, then it should be described. If not, would it be possible to obtain retroactively?
- Line 137: Why wasn’t the original post included in the analysis? To me, that seems an odd choice, so at least some brief rationale would be helpful to include.
- Line 143: It would be helpful to define corpora the first time it occurs
- Line 150: This may be my lack of social-media savvy showing, but the removal of bot comments seems like it could be a complex task that could bias the sample. As a result, it may merit brief elaboration.
- Line 152: An example of lemmatization would be helpful.
- I found the size of the behavior therapist comments to be a bit concerning. Building off of point 2 above, could it be helpful to include more general search terms for this population?
- Line 170: I am not familiar with this type of analysis, but, in any case, I think the statement that “residuals refer to statistical error” may be an oversimplification.
- Line 171: The authors describe requiring the models to meet held-out likelihood and residual criteria, but these criteria are never specified, which would seem essential to facilitating replication.
- Line 177: How were the top 10 representative comments determined? Given the importance of these comments in producing the ultimate label, more detail seems warranted.
- Line 181: How was agreement evaluated? Did raters have to provide identical labels?
- Line 186: Although I greatly appreciated the illustrative example, the substantial impact of viewing 10 additional comments on the label assigned was some cause for concern. The authors should describe why we should not expect that review of additional comments would lead to drastically different labels for other topics.
- Line 201: Given that edge width is based on the degree of positive correlation within the same comments and community membership is based on content similarities, it seems like it could be helpful to describe the difference/added value of the latter.
- Line 219: Is there a reason that “from behavior therapist subreddit members” is stipulated here but parallel statements are not made for the other groups?
Results
- To me, a bit more of a narrative description of different categories of topics that were identified could be helpful in guiding the reader through Figures 1-3. Otherwise, the task of reading through the entire list of topics to see which are common is a bit onerous. For example, in the autism topics the authors could note that some topics related to behavioral differences and experiences central to the diagnosis, support systems and strategies, experienced barriers or challenges, and areas of shared interest not directly related to autism.
- Relatedly, after reading further, I see that the authors get into this with the network analysis. Perhaps this suggests a more integrated review of the data could be helpful to make the theta data more useful while avoiding redundancy. Maybe the authors could present the topic data followed by the network analysis for each group, so the results of one analysis sort of flow into the next rather than jumping between groups.
- This is a bit more of an aesthetic point, but I’m a big fan of the viridis color palette in R, which the authors use in their network analysis graphs. I would encourage them to consider using this palette for the topic theta data. One option could be to assign color by expected frequency, but the authors could also assign color based on the results of the network analysis.
- Another aesthetic point is that the network analysis figures are quite cluttered and the topic labels makes it difficult to see the differences in color and width of the connecting lines, which are some of the most important outcomes of this process. The authors should consider alternative labeling methods. For example, perhaps each node can be superimposed with a letter and a color-coded key can provide the full topic label.
- It seems the network analysis is a rich source of data, so I found I odd that the authors exclusively focused on the community membership partitioning. It would be useful for the authors to incorporate some discussion of the edge or between-topic correlation data, especially given that this is difficult to evaluate visually in the crowded number of topics. For example, the authors could highlight the strongest connections for each group.
Discussion
- I appreciated the discussion of restricted interest as a potential strength rather than a limitation and the implications for the model/framework through which we approach autism. A recent survey (https://doi.org/10.1007/s42822-024-00191-4) evaluating the extent to which behavior analyst operate from a medical model (i.e., change the behavior) versus a social model (i.e., change the way others interact with the behavior) of disability may be useful to incorporate into this discussion. In my view, the results of this study suggest we have room to grow in operating from a more social model rather than defaulting/relying on a medical model. This could also be relevant to the future direction paragraph about revising intervention protocols.
- Line 400: The lengthy sentence beginning with “Aligned with current…” could benefit from revision
- Line 423: This last sentence seems to focus specifically on assessment and treatment of challenging behavior. If this was the focus of the discussion topics included in this analysis, then that is reasonable. However, if it is the case (which seems more likely) that topics also included discussion of skill acquisition procedures (e.g., early learning skills, social play skills, academic instruction), then examples related to these areas would be worth incorporating.
- One important point that is currently absent from the discussion is the fact that self-identified autistic individuals on Reddit are not representative of the full spectrum of individuals with autism. The needs and preferences of autistic individuals who may be more profoundly impacted by autism or may have less developed communication repertoires are likely distinct from those of the autistic individuals providing the comments on which this analysis was based. Accordingly, it would be inappropriate and even dangerous to assume that the clinical implications of such an analysis should alter the support services provided to a distinct population of autistic individuals. I would encourage the authors to incorporate a discussion of these points as well as alternative methods (e.g., concurrent chains preference assessment) for evaluating preferences and facilitating self-advocacy with autistic individuals that can’t meaningfully engage with Reddit.
Author Response
Comments 1: Line 31: It may be helpful to break down the diagnostic criteria into social and RRB domains and to presenting each component with a similar syntax (e.g., change “addressing the emergence of challenging behavior”)
Response 1: We appreciate this feedback, as we believe that the resulting changes (see Lines 29-31) are more aligned with the goals of this paper.
Comments 2: Line 35: Is mental health practitioners the most appropriate general label?
Response 2: Upon reflection, we feel that “behavior therapists” is the most appropriate general label to use, given that we use this terminology throughout the paper, and the term “mental health practitioners” is never used again throughout the text. We have replaced this term accordingly (Line 35).
Comments 3: Line 46: Although such terminology is pervasive, the authors may consider including terms like behavioral difference or tendency rather than symptom (especially given the goal of the paper).
Response 3: We have changed “symptom” to “behavioral tendency associated with autism,” as we agree that this change better aligns with the goal of this paper (Line 46).
Comments 4: Line 84: Rather than referencing Kapp et al specifically, it may be clearer to say previous qualitative work related to experiences of and supports for autistic individuals.
Response 4: Great suggestion. We have modified the sentence (Line 84).
Comments 5: In the introduction, the authors could do more to develop rationale for their use of subreddits, specifically. Specifically, it may be worth discussing whether or not certain communities may introduce bias in terms of what is said, how it I said, and by whom (i.e., why should we expect these samples to be sufficiently representative?).
Response 5: We have further developed our rationale for using subreddits as opposed to another social media site, and have chosen to put this information early in our Method section when first introducing Reddit (Lines 101-104).
Comments 6: Line 75: Details like unsupervised modeling, Latent Dirichlet Allocation, and Pennebaker's Linguistic Inquiry seem sufficiently complex to merit elaboration. Alternatively, if they are not essential to demonstrating the utility of this general approach, then they could be removed.
Response 6: To address this comment, we have removed the words/phrases “unsupervised,” “Latent Dirichlet Allocation,” and “Pennebaker’s Linguistic Inquiry” from our Introduction because they are not essential within the contexts that they are used (or are not used again later in the manuscript) and thus are not essential to the article. Instead, we now describe the Resnick and Resnick (2013) more generally, simply referring to their approach as “probabilistic topic modeling.” However, we have kept the text in which we define Latent Dirichlet Allocation in our Method section, where it becomes essential to a reader’s understanding of the article (Lines 76-78).
Comments 7: Line 93: Given the goal of the paper, it may be better to avoid “treatment of autistic individuals” and instead describe strategies for supporting members of this population or, at least, treating/intervening on their behavior.
Response 7: Thank you for providing constructive feedback that allows us to better align the language used in this paper with its overarching goals. Per your suggestion, we have modified “treatment of autistic individuals” to instead say “treatment of autistic individuals’ behavioral concerns,” to clarify that individual behaviors are the target of treatment rather than the individual themselves (Line 93).
Comments 8: Line 110: Missing a “c” in RedditExctactoR
Response 8: The missing “c” has been included (Line 114).
Comments 9: Line 119: The search terms for subreddits relevant to parents of autistic individuals seem especially general and likely to return many irrelevant results. Relatedly, the search terms are centric to behavior analysis seem more specific than the target population of “behavior therapists who serve in clinical settings with individuals with autism”.
Response 9: We appreciate you sharing your concerns about the possible over- and under-inclusivity of relevant search terms for this project. With regard to the inclusion of general terms such as “parent,” and related concerns that this may have returned irrelevant results, we refer to Lines 135-136 which states, “A keyword search of ‘autism’ was applied to ensure conversations in each subreddit concerned autism.” Thus, all comments collected through the use of the search term “parents” must have also included the keyword “autism” in order to remain in the data pool. Thus, we are confident that any potentially over-general search terms were similarly aided by this keyword inclusion. With regard to inclusion of more specific terms (e.g., behavior analyst, behavior technician), this functioned to ensure that our analyses would be focused on the types of practitioners that are most likely to regularly work closely with autistic individuals in a clinical setting.
Comments 10: Given the fundamental importance of identifying relevant and excluding irrelevant subreddits it would seem important to evaluate interrater agreement for application of the inclusion criteria. This is especially important given the subjective nature of the criteria. Was interrater agreement obtained. If so, then it should be described. If not, would it be possible to obtain retroactively?
Response 10: Explicit interrater agreement was not obtained for this step of data collection. We only obtained interrater agreement data for topic labels as part of this project. We conceptualized the process of identifying relevant and excluding irrelevant subreddits to be similar to the process conducted at the beginning of a literature review in which researchers discuss and agree upon the relevant terms, typically without collecting explicit interrater agreement data. Thus, this process was treated as a collaborative discussion in which all authors were verbally in agreement about which search terms to include.
Comments 11: Line 137: Why wasn’t the original post included in the analysis? To me, that seems an odd choice, so at least some brief rationale would be helpful to include.
Response 11: We have extended this sentence to include rationale for removing original posts from the analysis (Lines 141-143).
Comments 12: Line 143: It would be helpful to define corpora the first time it occurs
Response 12: We have moved our definition of corpora to occur after the first time it is used. Previously, it was defined after the second use in error (Line 149).
Comments 13: Line 150: This may be my lack of social-media savvy showing, but the removal of bot comments seems like it could be a complex task that could bias the sample. As a result, it may merit brief elaboration.
Response 13: We appreciate this comment, as we anticipate that some readers may also feel less familiar with the role of bots on social media sites such as Reddit. For our purposes, bot detection during pre-processing was limited to detecting repetitive, automatic responses to posts. These bot responses generally do not function to participate in or expand discussion and may instead promote products/services or alert a user that they have violated a guideline. We have included some brief elaboration of bot reply removal within this sentence in order to guard against concern that this pre-processing step may have unduly biased our sample (Line 156).
Comments 14: Line 152: An example of lemmatization would be helpful.
Response 14: We have added a sentence containing and example of lemmatization immediately after this sentence (Lines 159-160).
Comments 15: I found the size of the behavior therapist comments to be a bit concerning. Building off of point 2 above, could it be helpful to include more general search terms for this population?
Response 15: Although we appreciate the concern for the relatively smaller amount of comments in the behavior therapists group as compared to the other groups, we were okay with this difference given we were not making statistical comparisons across groups. In our limitations section of the Discussion, we address this by stating, “Finally, the number of comments we could extract from the behavior therapist subreddits was substantially smaller than the other two subreddits. As of April 3, 2023, a total of 61,337 individuals hold a BCBA certification, 5,520 individuals hold a BCaBA certification, and 136,113 individuals hold a Registered Behavior Technician (RBT) certification (Behavior Analyst Certification Board, 2023). We do not have data on the number of users whose comments contributed to the present study; however, given that we only obtained 3,163 comments from the behavior therapist subreddits, it can be assumed that our current sample represents only a tiny fraction of this community” (Lines 480-487). Unfortunately, we do not believe that including more general search terms is a potential solution to overcome this limitation, so we decided to not expand on this in our Discussion. However, if the AE believes we need to elaborate on potential solutions future researchers can consider, we are open to discussing this further.
Comments 16: Line 170: I am not familiar with this type of analysis, but, in any case, I think the statement that “residuals refer to statistical error” may be an oversimplification.
Response 16: We have replace the term “statistical error” with “the difference between predicted and observed outcomes” in service of more accurate description of residuals (Lines 177-178).
Comments 17: Line 171: The authors describe requiring the models to meet held-out likelihood and residual criteria, but these criteria are never specified, which would seem essential to facilitating replication.
Response 17: In this line of text, we note that model scores should meet the “criteria of higher held-out likelihood and lower residuals were obtained.” This is as specific as we can get with this description while remaining accurate, as, per Roberts et al. (2019), there are no exact criteria or target values for held-out likelihood or residuals. Rather, the decision is made in a relative manner consistent with our existing description. The values associated with held-out likelihood and residuals that we obtained for the present study would be irrelevant to a researcher replicating our methods and thus are not reported.
Comments 18: Line 177: How were the top 10 representative comments determined? Given the importance of these comments in producing the ultimate label, more detail seems warranted.
Response 18: We have included a statement within this sentence that clarifies how the top 10 most representative comments for each topic were calculated. Further, we associate this calculation with its common name (maximum-a-posteriori estimation) because, while that term is not particularly important for understanding our paper, a reader interested in replicating our methods may be aided in their search for additional literature by having some familiarity with this term (Lines 185-187).
Comments 19: Line 181: How was agreement evaluated? Did raters have to provide identical labels?
Response 19: We have included an additional sentence following the one noted above further describing how agreement was evaluated (Lines 191-193).
Comments 20: Line 186: Although I greatly appreciated the illustrative example, the substantial impact of viewing 10 additional comments on the label assigned was some cause for concern. The authors should describe why we should not expect that review of additional comments would lead to drastically different labels for other topics.
Response 20: We believe that some of your concern is caused by a lack of clarity in our chosen wording (e.g., referring to cases of rater disagreement as “disparate labels”). Upon reflection, using the term “disparate” feels rather extreme. Rather, any cases of disagreement were only slight, and could be reconciled by choosing a more comprehensive label that captured essential elements of both raters’ observations. The review of additional comments was simply a way to increase the sample of data under review to ensure that more comprehensive label was indeed an appropriate fit for the topic. As such, we have modified our language in this sentence to convey that any discrepancies that occurred were easily reconcilable through additional review, and that this review did not cause any drastic changes in labeling (Lines 194-196).
Comments 21: Line 201: Given that edge width is based on the degree of positive correlation within the same comments and community membership is based on content similarities, it seems like it could be helpful to describe the difference/added value of the latter.
Response 21: We have included a statement that community membership partitioning was done “to provide a broader layer of thematic analysis than what can be determined based on the correlation between two individual nodes alone,” in service of making this point clearer (Lines 227-229).
Comments 22: Line 219: Is there a reason that “from behavior therapist subreddit members” is stipulated here but parallel statements are not made for the other groups?
Response 22: There was not a reason for this stipulation; instead, it was an arbitrary inconsistency. We have removed this text such that parallel statements are made for each group (Lines 215-216).
Comments 23: To me, a bit more of a narrative description of different categories of topics that were identified could be helpful in guiding the reader through Figures 1-3. Otherwise, the task of reading through the entire list of topics to see which are common is a bit onerous. For example, in the autism topics the authors could note that some topics related to behavioral differences and experiences central to the diagnosis, support systems and strategies, experienced barriers or challenges, and areas of shared interest not directly related to autism.
Response 23: Because this comment is directly related to the subsequent comment, we have chosen to respond to both using the response to the subsequent comment (Comment 24).
Comments 24: Relatedly, after reading further, I see that the authors get into this with the network analysis. Perhaps this suggests a more integrated review of the data could be helpful to make the theta data more useful while avoiding redundancy. Maybe the authors could present the topic data followed by the network analysis for each group, so the results of one analysis sort of flow into the next rather than jumping between groups.
Response 24: We appreciate this suggestion regarding how to structure our paper in a way that feels more integrated. Given the nature of this research, we must either choose to organize our Results section by (1) type of analysis (topic proportion outcomes, network analysis outcomes) or (2) group under study (autistic individuals, parents/caregivers, behavior therapists). It is currently organized by the former, and we have considered shifting to the latter per your recommendation. However, ultimately, we suspect that the former is most friendly to readers given that it allows our Results section to mirror the structure of our Method section. As such, a reader may more easily follow exactly how we obtained the results that we did.
Comments 25: This is a bit more of an aesthetic point, but I’m a big fan of the viridis color palette in R, which the authors use in their network analysis graphs. I would encourage them to consider using this palette for the topic theta data. One option could be to assign color by expected frequency, but the authors could also assign color based on the results of the network analysis.
Response 25: We have updated all three of our topic proportions graphs to be visually represented using the viridis color palette (Figures 1-3; previously depicted in greyscale).
Comments 26: Another aesthetic point is that the network analysis figures are quite cluttered and the topic labels makes it difficult to see the differences in color and width of the connecting lines, which are some of the most important outcomes of this process. The authors should consider alternative labeling methods. For example, perhaps each node can be superimposed with a letter and a color-coded key can provide the full topic label.
Response 26: Thank you for this constructive comment and suggestion. In the process of writing this manuscript, we experimented with many ways to present these data visually, and considered taking an approach aligned with your suggestion. While doing so would indeed reduce clutter on the figures, we feel that their current presentation is the best approach. This is primarily because separating the labels from the nodes in favor of using a key would make the connections (or lack thereof) between topics more difficult for a reader to evaluate. We feel that allowing readers to quickly evaluate connections between topics was a worthwhile trade-off for having slightly “busier” figures, which is generally a difficult outcome to avoid when creating visual representations of network analyses.
Comments 27: It seems the network analysis is a rich source of data, so I found I odd that the authors exclusively focused on the community membership partitioning. It would be useful for the authors to incorporate some discussion of the edge or between-topic correlation data, especially given that this is difficult to evaluate visually in the crowded number of topics. For example, the authors could highlight the strongest connections for each group.
Response 27: We have incorporated discussion about the top three strongest correlations for each group at the end of each paragraph within our Network Analysis subsection of the Results section (Lines 294-296; 311-315; 326-329).
Comments 28: I appreciated the discussion of restricted interest as a potential strength rather than a limitation and the implications for the model/framework through which we approach autism. A recent survey (https://doi.org/10.1007/s42822-024-00191-4) evaluating the extent to which behavior analyst operate from a medical model (i.e., change the behavior) versus a social model (i.e., change the way others interact with the behavior) of disability may be useful to incorporate into this discussion. In my view, the results of this study suggest we have room to grow in operating from a more social model rather than defaulting/relying on a medical model. This could also be relevant to the future direction paragraph about revising intervention protocols.
Response 28: Thank you for this suggest and for making us aware of this important and timely project. We have added the following text and reference: “For additional discussion on this topic (i.e., the extent to which behavior analysts operate from a medical model versus a social model of disability), we direct the interested reader to the results of a recent survey on behavior analysts’ intervention strategies, which suggest we have room to improve in respecting behavioral diversity and prioritizing inclusion (Slanzi et al., 2024).” (Lines 398-403)
Comments 29: Line 400: The lengthy sentence beginning with “Aligned with current…” could benefit from revision
Response 29: We appreciate this comment and have revised the sentence to read as: “Aligned with current social network data literature predicting the dominance of vaccine misinformation on social media platforms in the next decade (Johnson et al., 2020), our analyses emphasize the necessity for evidence-based interventions against the potential spread of vaccine misinformation in parent subreddit communities.” (Lines 433-436)
Comments 30: Line 423: This last sentence seems to focus specifically on assessment and treatment of challenging behavior. If this was the focus of the discussion topics included in this analysis, then that is reasonable. However, if it is the case (which seems more likely) that topics also included discussion of skill acquisition procedures (e.g., early learning skills, social play skills, academic instruction), then examples related to these areas would be worth incorporating.
Response 30: Great catch. We added “…to teach skills and…” to ensure this recommendation is comprehensive. (Line 460)
Comments 31: One important point that is currently absent from the discussion is the fact that self-identified autistic individuals on Reddit are not representative of the full spectrum of individuals with autism. The needs and preferences of autistic individuals who may be more profoundly impacted by autism or may have less developed communication repertoires are likely distinct from those of the autistic individuals providing the comments on which this analysis was based. Accordingly, it would be inappropriate and even dangerous to assume that the clinical implications of such an analysis should alter the support services provided to a distinct population of autistic individuals. I would encourage the authors to incorporate a discussion of these points as well as alternative methods (e.g., concurrent chains preference assessment) for evaluating preferences and facilitating self-advocacy with autistic individuals that can’t meaningfully engage with Reddit.
Response 31: We are grateful for this comment and believe it is an excellent suggestion. We have added the following discussion and related references: “Importantly, it is vital to note that self-identified autistic individuals on Reddit are not representative of the full spectrum of individuals with autism. The needs and preferences of autistic individuals who may be more profoundly impacted by autism or may have less developed communication repertoires are likely distinct from those of the autistic individuals providing the comments on which this analysis was based (Wachtel et al., 2024). Accordingly, it would be inappropriate (and potentially dangerous) to assume that the clinical implications of such an analysis should alter the support services provided to a distinct population of autistic individuals. As such, other established methods for evaluating preferences and facilitating self-advocacy with autistic individuals who cannot meaningfully engage with Reddit must continue to be explored (e.g., concurrent chains preference assessments; Auten et al., 2024).” (Lines 484-494)
Reviewer 2 Report
Comments and Suggestions for Authors
Thank you for the opportunity to review the manuscript “An Exploratory Network Analysis of Discussion Topics About Autism Across Subreddit Communities” for consideration in Behavioral Sciences. The current manuscript explored Reddit subreddits and identified common threads within autistic groups, parents of autistic children, and behavioral therapists.
I must say, rarely am I left scrapping for comments to make to improve a paper after reviewing. This is one of those occasions. I found the manuscript incredibly well-written and engaging, and the study was designed nicely and examined thoroughly. It was a pleasure to read the manuscript, and I do not doubt that readers of this journal will also enjoy reading it. Therefore, my recommendation to the editor is that the article be accepted pending minor revisions, which you will notice below should not require too much work on the author’s end.
I want to reiterate how much joy this article was to read and acknowledge the level of detail and organization clearly presented in the writing. I certainly believe the paper is worth publication, and I hope readers can see it in print in Behavioral Sciences soon.
- Topic model and proportions could probably be explained sequentially in the method, as they seem to relate more closely than the Network Analyses. This may be particularly important for readability, as “topic proportions” is the last category described in the method but the first in the results.
- Given the large range of participants in each group, I found it somewhat difficult to interpret the results for the proportion of comments ranging between .02 and .06 for autistic individuals, .03 to .06 for parents, and .04 to .09 for behavior technicians. Perhaps including a frequency of comments and the proportion could be helpful.
- I appreciated the inclusion of the network analysis, particularly in its purpose of showing the larger number of topics discussed on Reddit pages, including autistic individuals, compared to behavior technicians. With that said, I found the heavily highlighted lines to be a bit (a) confusing and (b) potentially unnecessary. It is perhaps the case that I do not understand the explanation of a “positive correlation" as it relates to this analysis. Still, nonetheless, I did not find it added as much to the paper as the sheer number of topics and, of course, the topic analysis more broadly.
- I appreciated the discussion on vaccine hesitancy and misinformation among parents with autistic children. However, I also couldn’t help but notice the time frame of the current study. Given that these Reddit threads were identified during the COVID-19 pandemic and the COVID-19 vaccine was a common conversation, how did the authors ensure that conversations related to vaccines were in reference to autism rather than COVID-19?
Author Response
Comments 1: Topic model and proportions could probably be explained sequentially in the method, as they seem to relate more closely than the Network Analyses. This may be particularly important for readability, as “topic proportions” is the last category described in the method but the first in the results.
Response 1: We have reorganized our Method section such that the Topic Model section is directly followed by the Topic Proportions section, per your recommendation (previously, Network Analyses was between those two sections; Line 204).
Comments 2: Given the large range of participants in each group, I found it somewhat difficult to interpret the results for the proportion of comments ranging between .02 and .06 for autistic individuals, .03 to .06 for parents, and .04 to .09 for behavior technicians. Perhaps including a frequency of comments and the proportion could be helpful.
Response 2: Thank you for pointing out that the implications of comment proportions may be difficult for readers to interpret. To address this, we included additional verbiage outlining what this expected proportion means for the topic that made up the greatest expected proportion of comments for each group (Lines 242-243; 257-258; 273-275).
Comments 3: I appreciated the inclusion of the network analysis, particularly in its purpose of showing the larger number of topics discussed on Reddit pages, including autistic individuals, compared to behavior technicians. With that said, I found the heavily highlighted lines to be a bit (a) confusing and (b) potentially unnecessary. It is perhaps the case that I do not understand the explanation of a “positive correlation" as it relates to this analysis. Still, nonetheless, I did not find it added as much to the paper as the sheer number of topics and, of course, the topic analysis more broadly.
Response 3: We received another comment requesting descriptions of our findings relating to our strongest positive correlations within each group. In addressing that comment (see response to Comment 27 from Reviewer 1), we hope that the value of depicting these correlations visually becomes more clear.
Comments 4: I appreciated the discussion on vaccine hesitancy and misinformation among parents with autistic children. However, I also couldn’t help but notice the time frame of the current study. Given that these Reddit threads were identified during the COVID-19 pandemic and the COVID-19 vaccine was a common conversation, how did the authors ensure that conversations related to vaccines were in reference to autism rather than COVID-19?
Response 4: Thank you for this comment, as we agree that it is important to ensure that the data analyzed in this research is specifically related to autism. In service of this goal, you will find that our Method section (Data Extraction subsection, paragraph 3) states, “A keyword search of ‘autism’ was applied to ensure conversations in each subreddit concerned autism.” Thus, we confirm that all comments related to vaccines (and other topics) were related to autism in some way. It may be the case that some comments contained mention of both autism and COVID-19; however, we consider these to be relevant to the goal of this paper.